# Mechanisms of Rhizosphere Microbial Regulation on Ecosystem Multifunctionality Driven by Altitudinal Gradients in *Hylodesmum podocarpum*

**DOI:** 10.3390/biology14091126

**Published:** 2025-08-25

**Authors:** Kunlun Liang, Li Wang, Lili Nian, Mingyan Wang, Yang Li, Zhuxin Mao

**Affiliations:** 1Medical College, Huanghe Science and Technology University, Zhengzhou 450006, China; liangkunlun2008@163.com (K.L.); 200607130@hhstu.edu.cn (L.W.); wangmy201511@163.com (M.W.); 2Institute of Soil, Fertilizer and Water-Saving Agriculture, Gansu Academy of Agricultural Sciences, Lanzhou 730070, China; 18394797671@163.com; 3Xi’an Botanical Garden of Shaanxi Province, Institute of Botany of Shaanxi Province, Xi’an 710061, China; liyang@ms.xab.ac.cn

**Keywords:** *Hylodesmum podocarpum*, soil microorganisms, soil multifunctionality, community assembly

## Abstract

Mountain environments change dramatically with altitude, affecting the climate, soil, and the living organisms that depend on them. These changes can influence how well ecosystems perform multiple important functions, but the underlying biological mechanisms remain unclear. In this study, we investigated *Hylodesmum podocarpum*, a common legume in the Qinling Mountains of China, across four elevations from 896 to 1805 m. We focused on soil microbes living around the plant roots, known as the rhizosphere, because they play a key role in processes such as nutrient cycling, soil fertility, and plant growth. We found the highest microbial diversity at the highest elevation, where bacteria tended to work together while fungi tended to compete. Bacterial communities shifted from random patterns at lower sites to more predictable patterns at high elevation, while fungi remained random at all sites. Soil carbon and phosphorus were the strongest drivers of overall soil health, with bacteria mainly influencing carbon cycling and fungi affecting carbon, nitrogen, and phosphorus together. Our findings reveal that altitude shapes soil nutrients, which in turn influence microbial communities and ultimately determine how well mountain ecosystems function, providing insights for biodiversity conservation and sustainable land management.

## 1. Introduction

*Hylodesmum podocarpum* is a leguminous plant with good ecological adaptability and stress resistance, and it plays an important role in ecological restoration with its symbiotic nitrogen fixation ability of root nodules to improve soil fertility and promote the recovery of degraded ecosystems. Its well-developed root system contributes to soil and water conservation [1]. In addition, the plant may contain medicinal active ingredients (such as flavonoids or alkaloids), providing resources for traditional medicine or new drug development, and some species can also be served as high-quality forage or green manure to promote sustainable agriculture [2]. Studies have demonstrated that cultivating *Hylodesmum podocarpum* enhanced soil physicochemical properties and reshaped microbial community structure, consequently regulating ecosystem functioning [3]. Soil microorganisms are vital ecological agents, serving as key indicators of ecosystem health, soil quality, and land restoration progress. As important components of ecosystem processes, they drive critical functions such as plant productivity, nutrient cycling, and energy flow, ultimately underpinning ecosystem multifunctionality [4,5,6].

Ecosystem multifunctionality (EMF) is the ability of an ecosystem to provide multiple ecosystem functions and services simultaneously [7]. Soil carbon nutrient function, nitrogen nutrient function, and phosphorus nutrient function with soil nutrients as the core are the main factors restricting plant growth and ecological restoration in the Qinling Mountains [8]. Their nutrients not only act as essential reservoirs for plants, but also provide energy for soil microorganisms, regulating soil water and heat conditions and maintaining soil physical structure [9]. Therefore, soil multifunctionality assessment centered on soil nutrients is key to clarifying the soil ecological restoration of *Hylodesmum podocarpum* in the Qinling Mountains. Soil nutrient function is a key factor influencing soil microbial diversity and community structure [10]. Among environmental factors, altitude exhibits strong associations with microorganisms [11], primarily because altitude gradients regulate plant root secretions, thereby altering microbial habitats [12]. Therefore, we conducted an altitude gradient study in the Qinling Mountains, systematically examining how soil nutrient status affects rhizosphere microbial diversity of *Hylodesmum podocarpum* and its relationship with multiple ecosystem functions. Our study quantified a total of 20 parameters, encompassing soil properties, microbial diversity, nutrient cycling indicators, and plant productivity indicators. These included soil pH as a key physicochemical property, along with soil bacteria and fungi diversity. For nutrient cycling, we measured carbon-related indicators (total soil, stem, leaf, and soil sucrose activity), nitrogen-related indicators (total soil, stem, leaf, and soil urease activity), and phosphorus-related indicators (total soil, stem, leaf, soil alkaline phosphatase, and soil acidic phosphatase activity). Additionally, we assessed 17 primary productivity indicators of *Hylodesmum podocarpum*, including plant height, stem biomass, leaf biomass, and crown width.

Three hypotheses were proposed in this study: (1) Increasing altitude enhances the cooperative relationship among soil bacterial species but weakens the cooperative relationship among fungal species. (2) There are synergistic effects among soil carbon, nitrogen, and phosphorus nutrient functions, which promote each other and jointly enhance soil multifunctionality. (3) Altitude indirectly affects microbial diversity by regulating soil nutrient status, thereby driving changes in ecosystem functions. To verify these hypotheses, we employed high-throughput sequencing technology to characterize the rhizosphere microbial community diversity and structure of *Hylodesmum podocarpum* across altitude gradients. Molecular ecological network technology was combined to analyze the cooperative and competitive relationships among microbial genera, and the soil ecosystem multifunctionality and its main environmental drivers were evaluated based on the weighted mean method. The results will provide novel insights into soil microbial ecological processes at different altitudes in the Qinling Mountains and provide theoretical support for the sustainable utilization of *Hylodesmum podocarpum*.

## 2. Study Areas and Methods

### 2.1. Study Area

Zhashui County (108°50′–109°36′ E, 33°25′–33°56′ N) occupies a 2366.67 km^2^ area on the southern slope of the Qinling Mountains. Its geographical coordinates are shown in Figure 1. It exhibits characteristics of both northern and southern climate zones, with the northern part classified as warm temperate and the southeastern part as north subtropical. Overall, it has a monsoon climate transitioning from the subtropical to the warm temperate zone, with four distinct seasons, a mild climate, and abundant rainfall. The mean annual temperature is 10.3 °C, with a daily mean range of 7–20 °C. The mean annual precipitation is 1379.2 mm, with the rainy season concentrated from June to September. The unique natural conditions have made Zhashui County an excellent and optimal growing area for medicinal plants, earning its recognition as a “natural medicine storehouse”. The superior ecological environment and abundant resources favor the growth and reproduction of wild *Hylodesmum podocarpum*. Notably, Huanghualing in Yingpan Town serves as the core germplasm distribution center for *Hylodesmum podocarpum*. It boasts abundant resources and excellent germplasm, making it have significant conservation and utilization value.

### 2.2. Sample Collection

The experiment began on 6 July 2024, in the Huanghualing area of Yingpan Town, Zhashui County, on the southern slope of the Qinling Mountains, with four different altitude gradients set up, namely 896 m (HB1), 1407 m (HB2), 1597 m (HB3), and 1805 m (HB4). At each altitude gradient, five Schisandra chinensis plants were randomly selected and soil samples of *Hylodesmum podocarpum* from their rhizosphere were collected using the “stripping separation method”. The method was as follows: dig out the Schisandra chinensis plants with soil along with the roots, gently shake off the large pieces of soil outside the roots (considered as non-rhizosphere soil), forcefully shake off the soil attached to the surface of the roots, and quickly collect them in a clean plastic bag as rhizosphere soil samples. After thorough mixing, the collected soil samples were divided into three parts: one part was placed in a sterile centrifuge tube and stored in a refrigerator at −80 °C for extracting total DNA from the soil; another part was kept in a refrigerator at 4 °C for soil enzyme activity analysis; and the remaining part was air-dried, ground, and sieved for determining the basic physical and chemical properties of the soil, including pH, total carbon (TC), total nitrogen (TN), total phosphorus (TP), and total potassium (TK). In addition, samples of *Hylodesmum podocarpum* plants were collected simultaneously at each altitude gradient to determine the contents of nutrients such as nitrogen, phosphorus, and potassium.

### 2.3. Soil Physicochemical Analysis

Soil pH was measured using a pH meter (Mettler Toledo, Greifensee, Switzerland) at a soil-to-water ratio of 1:2.5. Soil and plant carbon contents were determined using the potassium dichromate external heating method [13]. Soil total nitrogen and plant nitrogen contents were measured using the semi-micro-Kjeldahl method [14], while soil total phosphorus and plant phosphorus contents were determined by the ammonium molybdate spectrophotometric method [15]. Soil sucrase activity was determined using the 3,5-dinitrosalicylic acid colorimetric method [16], soil urease activity was measured by the indophenol blue colorimetric method [17], and soil alkaline phosphatase and soil acid phosphatase activities were measured by the p-nitrophenyl phosphate method [18].

### 2.4. Soil DNA Extraction, High-Throughput Sequencing, and Bioinformatics Analysis

Total soil DNA was extracted from 0.25 g of fresh soil in each sample using the PowerSoil DNA Isolation Kit (MO BIO Laboratories, Carlsbad, CA, USA) in accordance with standardized operating procedures. To minimize operational error, each soil sample was repeatedly extracted three times, and the resulting DNA extracts were combined for subsequent analysis. The extracted DNA was tested for integrity by 1% agarose gel electrophoresis, and its concentration and purity were determined using the NanoDrop ND-2000c ultraviolet–visible spectrophotometer (NanoDrop Technologies, DE, USA). The amplification of the V3-V4 region of the bacterial 16S rRNA gene used primers for 515F (5′-GTGCCAGGCGCCGCGCGGTA-3′) and 907R (5′-CCGTCAATTCCTTGAGTTT-3′), and the amplification of the ITS region in fungi used primers for ITS5-1737F (5′-GGAAGTAAAAGTCGTAACAAGG-3′) and ITS2-2043R (5′-GCTGCGTTCTTCATCGATGC-3′) [16]. The PCR amplification products were purified and sent to Shanghai Meiji Biomedical Technology Co., Ltd.,(Shanghai, China) for high-throughput sequencing using the Illumina MiSeq platform.

The diversity of soil microbial communities at different altitudes was analyzed using the “vegan” and “picante” software packages in R language. α diversity includes the Chao1 index, ACE index, Shannon index, and Simpson index, all calculated based on OTU levels, used to measure the richness and diversity of microbial communities. β diversity assesses the dissimilarity of microbial communities at different altitudes through the Bray–Curtis distance, revealing variations in community structure.

### 2.5. Determination of Ecosystem Multifunctionality

A total of 17 ecosystem function (EF) indicators were measured in this study, and these indicators were divided into four ecosystem function groups: (1) functions related to nutrients and storage of soil carbon (EF-C: total soil carbon, total stem carbon, total leaf carbon, and soil sucrase); (2) functions related to nutrients and storage of soil nitrogen (EF-N: total soil nitrogen, total stem nitrogen, total leaf nitrogen, and soil urease); (3) functions related to nutrients and storage of soil phosphorus (EF-P: total soil phosphorus, total stem phosphorus, total leaf phosphorus, soil alkaline phosphatase, and acidic phosphatase); (4) soil ecosystem multifunctionality (EMF: including all 17 ecosystem function indicators). These functional indicators were chosen because they can reflect and regulate key ecological processes in the ecosystem where *Hylodesmum podocarpum* grow and are widely used in the study of ecosystem function and multifunctionality [19,20,21].

The ecosystem multifunctionality index was calculated using the mean method [7,22]. First, the 17 ecological function indicators were standardized.$$f_{ij}=\frac{x_{ij}-{min}_{ij}}{{max}_{ij}-{min}_{ij}}$$
where $f_{ij}$ is the standardized value of the jth ecosystem function variable in plot *i*, $x_{ij}$ is the actual measured value of the jth ecosystem function variable in plot *i*, ${min}_{ij}$ is the minimum value of the jth ecosystem function variable among all plots of the same factor, and ${max}_{ij}$ is the maximum value of the jth ecosystem function variable among all plots of the same factor.

The ecosystem function index EF is calculated by the single function method:$${EF}_{ij}=\sum_{j}^{n}\frac{f_{ij}}{n}$$

The ecosystem versatility index EMF is calculated using the average method:$${EMF}_{i}=\frac{1}{N}\sum_{1}^{N}f_{ij}$$
where ${EF}_{ij}$ is the functional index of the jth function of plot *i*; *n* is the number of ecosystem variable indicators contained in the function; EMF is the ecosystem multifunctionality index of the plot, calculated based on the standardized average of all variable indicators of the plot; and *N* is the number of all ecosystem functions contained in the plot i.

### 2.6. Calculation of Soil Element and Enzyme Metering Ratios

In this study, soil element ratios are used to represent soil nutrient limits, and vector angles (VAs) and vector lengths (VLs) calculated by soil enzyme stoichiometric ratios are used to represent soil microbial element limits [23]. The calculation formula is as follows:VL = SQRT [(CEs: PEs)^2^ + (CEs: NEs)^2^]VA = Degrees [Atan2(CEs: PEs, CEs: NEs)]
where CES represents the activity of soil carbon-degrading enzymes (soil sucrase), NES represents the activity of soil nitrogen-degrading enzymes (urease), and PES represents the activity of soil phosphorus-degrading enzymes (alkaline phosphatase and acid phosphatase). The longer the VL, the stronger the carbon restriction of the microorganism. VAs less than 45° and greater than 45° indicate nitrogen and phosphorus restriction of the microorganism, respectively.

### 2.7. Data Processing

This study used the Majorbio cloud platform (https://cloud.majorbio.com/, accessed on 18 May 2025) of different elevation gradients of microbial community diversity and structure analysis. Data statistical analysis of soil physicochemical properties was performed using Excel 2016, and graphs of the relative abundance of microorganisms at the genus level were plotted. For statistical analysis, one-way ANOVA was performed using SPSS 25.0, and the significance of differences was tested using the multiple comparison method (LSD method, *p* = 0.05). Molecular ecological networks were calculated and constructed using R language and visualized using Gephi software 9.2. The relationship between microbial genus levels and soil physicochemical factors was analyzed using Canoco5 for redundancy analysis (RDA). Community assembly processes were calculated using R language, box plots were drawn using Origin 2021, and the final charts were beautified and formatted using Adobe Illustrator 2020. In addition, the partial least squares path model (PLS-PM) was used to investigate the path of the impact of altitude on soil multifunctionality, and modeling analysis was conducted using the plspm package in R language.

## 3. Results and Analysis

### 3.1. Soil and Plant Physical and Chemical Properties

There are significant differences in nutrient indicators between soil and plants at different altitudes. Among them, the soil total carbon, soil total nitrogen, and leaf total phosphorus at an altitude of 1407 m (HB2) were significantly higher than those at other altitude gradients (*p* < 0.05), while the soil total potassium, stem total nitrogen, and stem total phosphorus at an altitude of 1597 m (HB3) were significantly higher than those at other altitude gradients (*p* < 0.05, Figure 2). Although the differences in soil urease, catalase, alkaline phosphatase, superoxide dismutase, sucrase, and acid phosphatase were not significant at different altitudes, the indicators showed a certain trend of variation at different gradients: superoxide dismutase and acid phosphatase activities were highest at 896 m (HB1); catalase, alkaline phosphatase, and sucrase activities were higher at HB2; and urease activity was higher at HB3 compared to other gradients, indicating that altitude changes influenced the distribution pattern of soil enzyme activities to some extent.

### 3.2. Soil Microbial Community Structure Characteristics

High-throughput sequencing analysis of the soil microbial community (Figure 3) revealed that *Actinobacteriota*, *Acidobacteriota,* and *Proteobacteria* were the dominant phyla of bacteria. The total relative abundance of the three at different altitude gradients was 78.97% (HB1, 896 m), 76.33% (HB2, 1407 m), 77.82% (HB3, 1597 m), and 78.35% (HB4, 1805 m), respectively. This shows a trend of decreasing first and then increasing with the increase in altitude. Specifically, the relative abundance of *Acidobacteriota* was lowest at HB4 (14.55%) and highest at HB2 (22.38%). The relative abundances of *Proteobacteria* and *Actinobacteriota* were highest at HB4 at 29.71% and 34.09%, respectively, and lowest at HB2 at 25.69% and 28.26%, respectively. In the fungal community, *Ascomycota*, *Basidiomycota,* and *Mortierellomycota* were the dominant phyla. With the increase in altitude, the total relative abundance of the three increased in sequence to 87.73% (HB1), 92.30% (HB2), 95.13% (HB3), and 95.73% (HB4), indicating a continuous upward trend in the dominant phylum of the fungal community. Further analysis showed that *Ascomycota* had the lowest relative abundance (53.70%) at HB3 and the highest (69.99%) at HB2; *Mortierellomycota* had the highest relative abundance (21.05%) at HB1 and the lowest (4.05%) at HB2.

To investigate the effects of different altitude gradients on the structure of soil bacterial and fungal communities, principal coordinate analysis (PCA, Figure 4) was performed based on the Bray–Curtis distance matrix in this study. The results showed that the bacterial and fungal communities were clearly divided into two groups on the principal coordinate axis: one group was HB4 (at an altitude of 1805 m), and the other group was HB1, HB2, and HB3 (at altitudes of 896 m, 1407 m, and 1597 m, respectively). Among them, HB4 was more distant from the soil bacterial and fungal community structures of the other altitude gradients, indicating that there were differences in the soil bacterial and fungal community structures at different altitudes. This result was further verified by ANOSIM analysis, which showed significant differences in bacterial and fungal communities at different altitudes (*p* < 0.05). Significant differences in bacterial and fungal diversity and richness were observed among treatments (Figure 5). For bacteria, the Shannon and Chao1 indices were lower in HB3 than in the other treatments, whereas they were highest in HB4. For fungi, the Shannon and Chao1 indices were highest in HB1 and lowest in HB2. Overall, microbial diversity and richness were highest in HB4 and lowest in HB3.

Significant differences were observed in soil microbial co-occurrence network characteristics across different altitudes (Figure 6, Table 1). The results of the soil bacterial co-occurrence network analysis indicated that at an altitude of 1597 m (HB3), the number of network connections (5465), average weighted (40.073), and graph density (0.147) all reached their peak, suggesting that the bacterial co-occurrence network at HB3 was more complex, with closer and more complex interactions among species. In addition, at an altitude of 1805 m (HB3), the proportion of positively correlated edges was highest, indicating that synergies were more significant in the bacterial community at this altitude gradient and that there were stronger cooperative relationships among species. The analysis of the fungal co-occurrence network showed that at HB4, the number of connections (4581), average weighted degree (36.502), and graph density (0.146) of the fungal network were also highest, indicating that the fungal network structure was the most complex and interspecies interactions were more active at this altitude. However, unlike the bacterial community, the proportion of positively correlated edges in the fungal network was lowest at HB4 and highest at HB2, which may reflect the weakening of cooperative relationships and the strengthening of competitive interactions among species in the fungal community as the altitude increased.

To explore the ecological mechanisms driving the differences in soil microbial community structure in *Hylodesmum podocarpum* at different altitude gradients, the zero-model analysis method was used to study the assembly process of the microbial community (Figure 7). The results showed that the assembly mechanism of the soil bacterial community was dominated by random processes at altitudes of 896 m (HB1), 1407 m (HB2), and 1597 m (HB3), while it was dominated by deterministic processes at 1805 m (HB4). The formation of bacterial communities was mainly driven by drift and other ecological processes in the random process, as well as heterogeneous selection in the deterministic process. As altitude increases, the contribution of drift and other random ecological processes decreases, while the influence of heterogeneous selection increases, indicating that environmental selection pressure increases at high altitudes and plays a more significant decisive role in bacterial community structure. The zero-model analysis of soil fungal communities showed that at different altitude gradients, the assembly of the communities was dominated by random processes, mainly determined by the diffusion limitation, drift, and other random processes. As altitude increases, the effect of diffusion restriction in the fungal community gradually weakens, while the effect of homogeneous diffusion increases.

### 3.3. Soil Ecosystem Multifunctionality

Soil stoichiometric and enzymatic stoichiometric characteristics are shown in Figure 8. Soil C/P and N/P ratios vary significantly at different altitudes (*p* < 0.05), while C/N ratios do not vary significantly at different altitudes. The C/N ratio at HB4 was higher than that at other altitude gradients, while the C/P and N/P ratios at HB2 were significantly higher than those at other altitude gradients (*p* < 0.05). This indicates that the soil at HB2 is more restricted by phosphorus. Microbial nutrient restriction was further investigated through vector analysis of soil enzyme activity. The longer the enzyme vector length (VL), the greater the degree of carbon restriction on microorganisms. An enzyme vector angle (VA) greater than 45° indicates that soil microorganisms are mainly restricted by phosphorus.

The results showed that the VL in HB2 was higher than in other treatments, indicating stronger carbon limitation at HB2. In addition, the VA values for all treatments were below 45°, suggesting that microorganisms in all altitude gradients were subject to phosphorus limitation.

A one-way analysis of variance was conducted on the single function of the *Hylodesmum podocarpum* ecosystem at different altitude gradients, and the results are shown in Figure 9. The carbon nutrient function, phosphorus nutrient function, and ecosystem multifunctionality of the rhizosphere soil of *Hylodesmum podocarpum* at different altitudes were all higher at HB2 than at other altitude gradients; soil nitrogen nutrient function was higher at HB3 than at other altitude gradients; and *Hylodesmum podocarpum* productivity was higher at HB4 than at other altitude gradients. Among them, there was no significant difference in productivity and ecosystem multifunctionality among the different altitude gradients, but there were significant differences in soil carbon nutrient function, soil phosphorus nutrient function, and soil nitrogen nutrient function among the different altitude gradients (*p* < 0.05). There was a synergistic relationship among soil carbon nutrient function, soil phosphorus nutrient function, and soil nitrogen nutrient function. The increase in each function promoted the improvement in soil multifunctionality, and the correlation coefficients of soil multifunctionality with each single function were 0.78, 0.77, and 0.47, respectively.

To reveal the mechanism by which altitude gradient affects ecosystem multifunctionality (Figure 10), this study constructed a structural equation model (SEM) to clarify the path relationship between environmental factors, microbial community structure, and ecological function. The results showed that altitude had a significant positive effect on both nitrogen and phosphorus nutrient functions in the soil (*p* < 0.05). Soil pH had a significant positive effect on soil nitrogen nutrient function (*p* < 0.05), but a significant negative effect on carbon nutrient function (*p* < 0.05). Fungal community structure showed significant negative effects on soil nitrogen nutrient function, carbon nutrient function, phosphorus nutrient function, and ecosystem multifunctionality (*p* < 0.05). In contrast, bacterial community structure showed a significant positive effect on soil carbon nutrient function (*p* < 0.05).

### 3.4. Relationship Between Soil Microbial Communities and Ecological Functions

To investigate the dominant ecological function factors driving changes in the composition of soil bacterial and fungal communities, this study conducted a redundancy analysis (RDA) using the composition at the phylum level of bacterial and fungal communities as response variables and nutrient function indicators as explanatory variables (Figure 11). The results showed significant differences in the responses of the main groups of soil bacterial communities to different nutrient functional factors. *Chloroflexi*, *Bacteroidota*, *Gemmatimonadota,* and *Myxococcota* were positively correlated with soil nitrogen nutrient function (EF-N), phosphorus nutrient function (EF-P), and ecosystem multifunctionality (EMF). *Actinobacteriota*, *Verrucomicrobiota,* and *Planctomycetota* were positively correlated with carbon nutrient function (EF-C). RDA of soil fungal communities showed that Basidiomycota and Chytridiomycota were positively correlated with soil pH, EF-N, EF-P, and EMF, while *Rozellomycota* was positively correlated with EF-C. Combined with the RDA results, nitrogen nutrient function (EF-N) showed strong explanatory power in the composition changes of bacterial and fungal communities and was the main nutrient function factor affecting the structure of microbial communities.

## 4. Discussion

### 4.1. Effects of Altitude Gradient on Soil Microbial Community Characteristics of Hylodesmum podocarpum

Our high-throughput sequencing analysis of *hylodesmum podocarpum* rhizosphere soil across altitudes gradients in the Qinling Mountains revealed distinct microbial community patterns. In this study, *Actinobacteriota*, *Acidobacteriota,* and *Proteobacteria* were identified as the dominant bacterial phyla, and the total relative abundance of the dominant bacterial phyla decreased first and then increased with altitude. *Ascomycota*, *Basidiomycota,* and *Mortierellomycota* were the dominant phyla of fungi, and the total relative abundance of the dominant phyla of fungi increased with altitude. These patterns likely reflect fundamental ecological adaptations. The warmer temperatures and richer nutrient availability at lower elevations favor bacterial growth, while fungi such as *Ascomycota* and *Basidiomycota* could tolerate colder and drier environments while having a competitive advantage in the cooler higher elevations [24,25,26]. Furthermore, our results showed that microbial diversity and richness were higher in the HB4 treatment than in the other treatments, whereas they were lower in the HB3 treatment. The higher diversity in HB4 may be attributed to its relatively low level of human disturbance, more favorable climatic conditions (e.g., lower temperatures and higher humidity that promote organic matter accumulation), and the resulting higher habitat heterogeneity [27,28]. In addition, stronger ultraviolet radiation at higher elevations may have selected for unique stress-resistant microbial taxa. In contrast, the lower diversity observed in HB3 may be related to the moderate environmental stress typical of its altitudinal zone, potentially greater human disturbance, and relatively limited resource availability [29]. Thus, altitudinal differences, by regulating disturbance intensity, climatic patterns, resource availability, and habitat complexity, ultimately led to the pronounced divergence in microbial diversity and richness between the HB3 and HB4 treatments.

Molecular ecological networks analysis can not only reflect the interactions among different groups within a community, but also be used to assess the complexity of the target community, and have been successfully applied to analyze the impact of environmental characteristics on microbial communities [30]. Our results showed soil–microbial co-occurrence networks vary significantly across different altitude gradients. Notably, bacterial communities at higher elevations exhibited stronger cooperative relationships, while fungal communities exhibited increasing competitive relationships. This divergence likely reflected differential adaptive strategies under high-altitude stress conditions. Bacterial communities would enhance their survival competitiveness by strengthening their cooperative relationships, while fungal communities would share resources through cooperative relationships, such as mycelial network interactions. The survival competitiveness of the entire community is enhanced [31,32]. In addition, we found that in the HB3 treatment, soil bacterial co-occurrence networks were more complex, with intricate interactions among species, whereas in the HB4 treatment, fungal co-occurrence networks exhibited higher complexity. In HB3, seasonal disturbances combined with moderate temperatures drove the rapid input of labile resources, enabling fast-responding bacteria to secure resources via dense competition–cooperation networks, thereby forming a complex bacterial network [33,34]. In contrast, in HB4, the low-temperature, high-humidity environment suppressed bacterial activity but favored fungal hyphal expansion. Together with the input of litter enriched in lignin and cellulose, these conditions likely drove fungi to evolve highly modular, functionally specialized networks, further enhancing fungal network connectivity and ultimately constructing a complex fungal interaction system [35]. In conclusion, different altitudes affect the structure and function of microbial communities by altering factors such as soil physicochemical properties, nutrient cycling, plant residue composition, and soil microenvironment, thereby influencing interactions among species.

To explore the drivers of altitude variation in *hylodesmum podocarpum* rhizosphere microbial communities, we analyzed the intrinsic assembly mechanism of microbial community distribution patterns based on the null model. Bacterial community assembly was dominated by random assembly processes at HB1, HB2, and HB3, while it was dominated by deterministic processes at HB4. Along the elevational gradient, drift and other ecological processes decreased as heterogeneous selection increased. This shift may reflect environmental filtering intensification (temperature drops, moisture decreases, and ultraviolet radiation intensifies), leading to an increase in the dominance of deterministic processes (especially heterogeneous selection) at HB4 and an increase in the harshness of high-altitude habitats and environmental heterogeneity, resulting in a strong screening of microbial groups adapted to specific conditions. Consequently, random processes (drift and other ecological processes) become less influential in community assembly at upper elevations. In contrast, the environmental conditions at lower altitudes are relatively mild and the assembly of microbial communities is more driven by random processes [36,37]. Fungal communities showed different patterns, being primarily assembled by random assembly processes at all elevations, displaying reduced diffusion limits and increased homogeneous diffusion with altitude. These patterns indicated that altitude increases create more uncertain conditions for fungal growth, while also allowing certain adaptive fungal groups to spread frequently across habitats, increasing community similarity among different sites [38].

### 4.2. The Effect of Altitude Gradient on the Multifunctionality of the Soil Ecosystem of Hylodesmum podocarpum

Altitude exerts profound effects on plant growth, soil properties, and microbial communities, ultimately influencing forest ecosystem stability and multifunctionality [27]. Our results revealed distinct altitudinal patterns in nutrient cycling. In this study, soil carbon nutrient function, soil phosphorus nutrient function, and ecosystem multifunctionality were all higher at HB2 than at other altitude gradients, and soil nitrogen nutrient function was higher at HB3 than at other altitude gradients. These patterns might reflect optimal environmental conditions at HB2, where moderate temperature and moisture levels support higher vegetation productivity and litter input. This promoted soil organic carbon accumulation and microbial activity, thereby enhancing the carbon-phosphorus cycle efficiency. In addition, the environmental conditions at HB2 are closer to the optimal range of microbial metabolism and plant–soil interaction, which supports higher ecosystem multifunctionality [39]. In contrast, the cooler HB3 environment may reduce nitrogen loss by inhibiting nitrification. The restricted growth of plants at higher altitudes reduces the competition for nitrogen absorption, resulting in a relative enrichment of available nitrogen in the soil [40]. In addition, soil nutrient functions are composed of soil physicochemical indicators, and the effect of altitude on soil physicochemical indicators is ultimately manifested as the effect on the ecosystem functions formed by the corresponding indicators. Notably, our results showed synergistic interactions among soil carbon nutrient function, soil phosphorus nutrient function, and soil nitrogen nutrient function. These nutrient functions interact synergistically to enhance soil multifunctionality, with soil carbon nutrient function and phosphorus nutrient function playing particularly important roles. This synergy may occur because soil organic carbon serves as an energy source for microbial activity and nutrient cycling, significantly boosting the metabolic activity of microbial communities [41]. Higher carbon availability not only directly supports the growth of decomposers, but also accelerates organic mineralization through excitation effects, indirectly releasing bound phosphorus and nitrogen. This “carbon activation–nutrient release” mechanism forms the foundation for enhancement in multifunctionality [42].

This study investigated the mediating role of soil–microbial factors in altitude gradients on ecosystem multifunctionality in forest ecosystems. The results showed a dual pathway of altitude influence. On one hand, altitude can directly affect forest ecosystem multifunctionality as a weak positive effect, because environmental gradients such as temperature, precipitation, and ultraviolet radiation remained within the functional compensation thresholds [43]. On the other hand, altitude can indirectly affect forest ecosystem multifunctionality through fungal communities as a significant negative effect. Stress-tolerant ectomycorrhizal fungi in high-altitude environments compete for carbon through plant symbiosis, suppressing saprophytic fungus activity and impairing C-N-P cycling efficiency [44]. This is consistent with the significant negative effects of fungal communities on soil nutrient function in this study. These patterns could be further explained by specific microbial-mediated mechanisms. Firstly, soil bacterial communities enhanced carbon nutrient function, possibly through dynamic regulation of carbon use efficiency under resource limitation, because bacteria can balance decomposition and retention processes [45]. Secondly, soil pH has important effects. The acidic soil suppressed soil carbon storage by inhibiting microbial activity, reducing organic matter decomposition, and inducing Al/Fe toxicity in plants [46]. Neutral-alkaline soils (pH 7.01–7.57) promoted nitrogen cycling through two synergistic pathways: stimulating ammonia-oxidizing bacteria and nitrogen-fixing bacteria, and enhancing urease activity within its optimal pH range (6.0–7.5) [47,48]. Importantly, these microbial mechanisms collectively explain why fungal-dominated high-altitude sites showed reduced multifunctionality despite comparable nutrient inputs.

This study revealed the significant effects of altitude on soil microbial diversity and ecosystem multifunctionality of *Hylodesmum podocarpum* habitats in the Qinling Mountains. The results indicate that moderate altitudes have a promotive effect, whereas excessively high altitudes exert negative impacts. To better understand the mechanisms underlying the long-term maintenance of soil ecosystem functions across different altitudinal gradients, future research should focus on the structure and function of the soil micro-food web and its integrative regulatory role in ecosystem multifunctionality.

## 5. Conclusions

This study demonstrates that altitude gradients significantly influence microbial communities and ecosystem multifunctionality in the *Hylodesmum podocarpum* rhizosphere soils of the Qinling Mountains. Soil pH significantly affects soil carbon nutrient function and nitrogen nutrient function, with pH having a negative effect on carbon nutrient function and a positive effect on nitrogen nutrient function. Bacterial community structure has a significant positive effect on soil carbon nutrient function, while fungal community structure has a significant negative effect on nutrient function and ecosystem multifunctionality. In addition, there were significant synergistic effects among the three nutrient functions of soil carbon, nitrogen, and phosphorus, and the mutual promotion of these functions significantly enhances soil multifunctionality centered on nutrient functions. Among them, carbon nutrient function and phosphorus nutrient function contribute relatively more to the multifunctionality. To sum up, this study systematically revealed the influence mechanism of altitude gradient on the soil microbial community structure and ecosystem multifunctionality of *Hylodesmum podocarpum* from the perspectives of microbial assembly mechanisms, diversity dimensions, and ecological function pathways, providing a scientific basis and theoretical support for a deeper understanding of microbial ecological processes and their functional maintenance mechanisms in mountain ecosystems.

## Figures and Tables

**Figure 1 biology-14-01126-f001:**
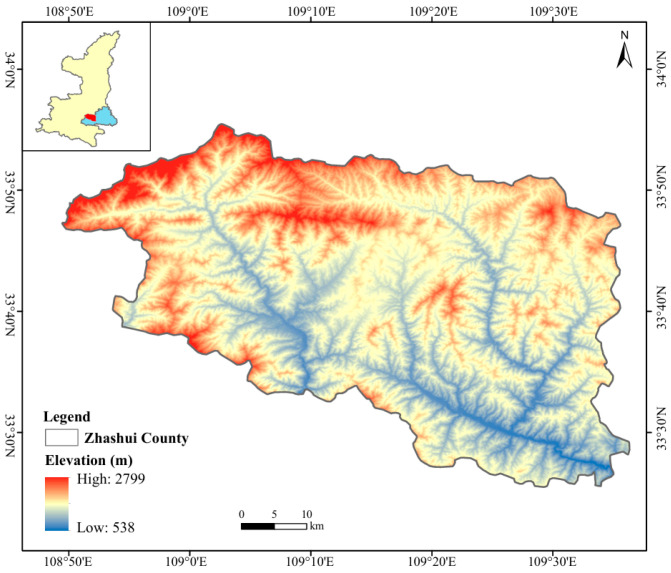
Location of study area.

**Figure 2 biology-14-01126-f002:**
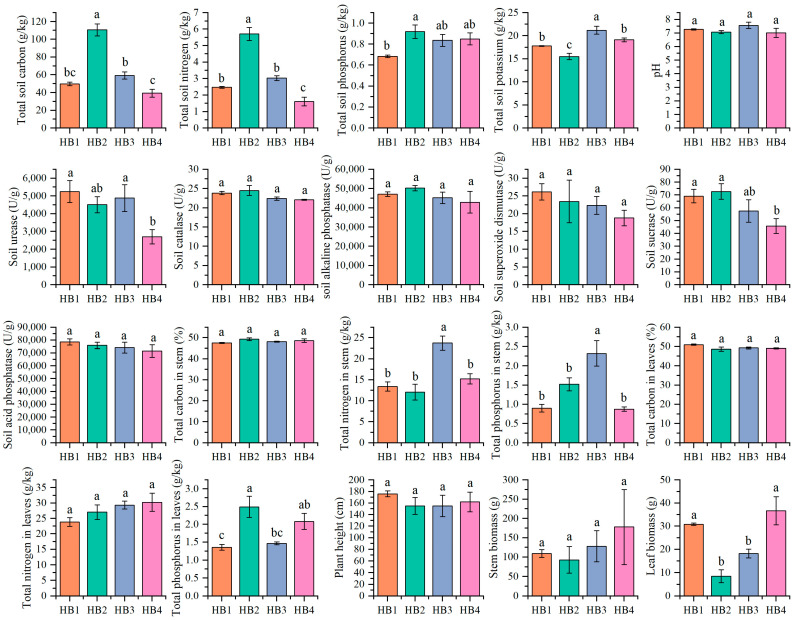
Plant physiological and ecological traits and soil physicochemical properties. Note: different letters indicate significant differences among different altitude gradients (*p* < 0.05). HB1—896 m; HB2—1407 m; HB3—1597 m; and HB4—1805 m; the same as below.

**Figure 3 biology-14-01126-f003:**
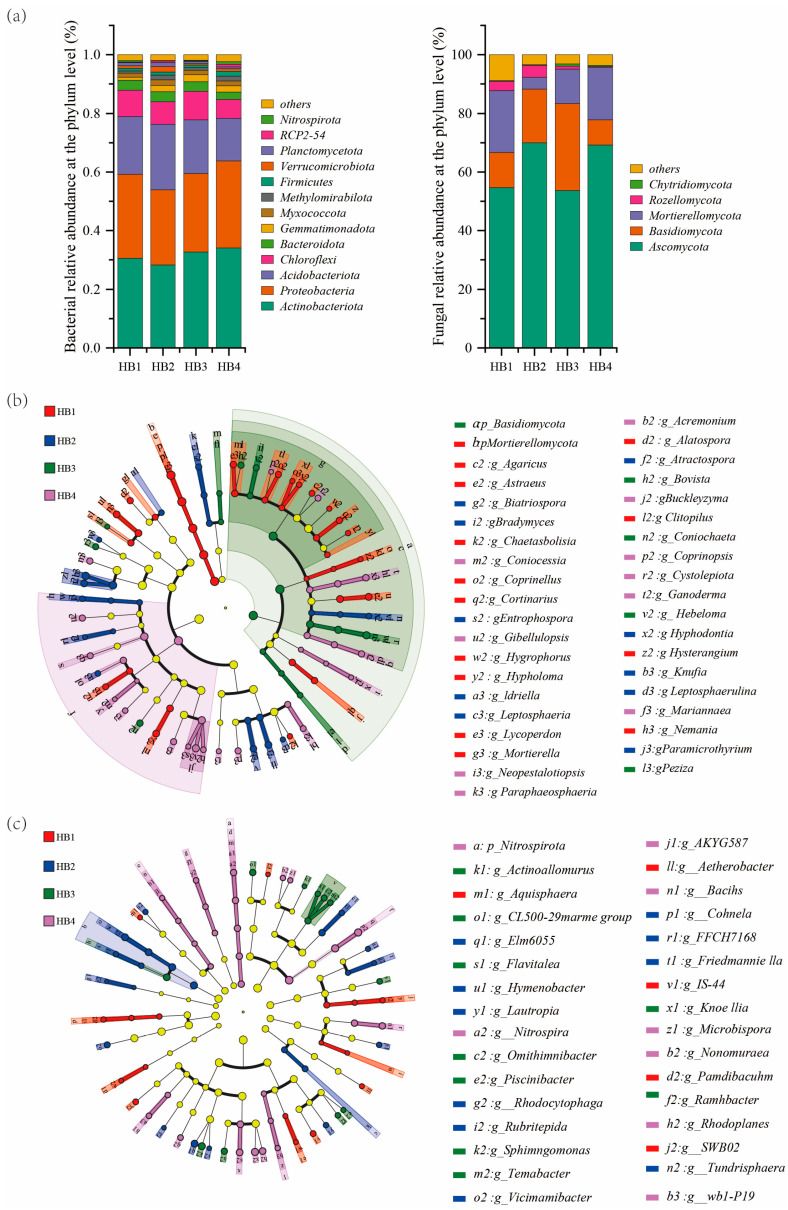
Relative abundance of bacterial and fungal phyla at different altitude gradients. Note: (**a**) microbial community composition; (**b**) differential analysis of bacterial communities; (**c**) fungal community differential analysis.

**Figure 4 biology-14-01126-f004:**
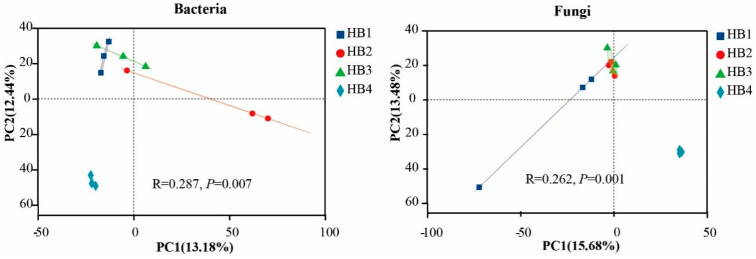
Principal component analysis (PCA) of bacterial and fungal communities at different altitude gradients.

**Figure 5 biology-14-01126-f005:**
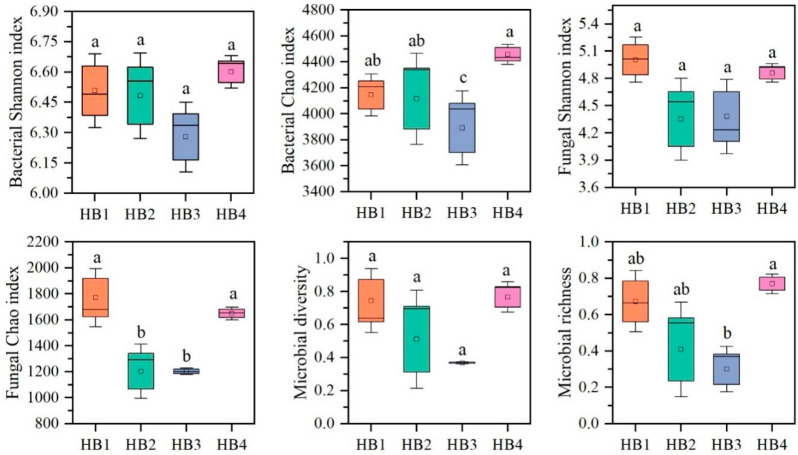
Diversity indices of soil microbial communities. Note: different letters indicate significant differences between treatments (*p* < 0.05).

**Figure 6 biology-14-01126-f006:**
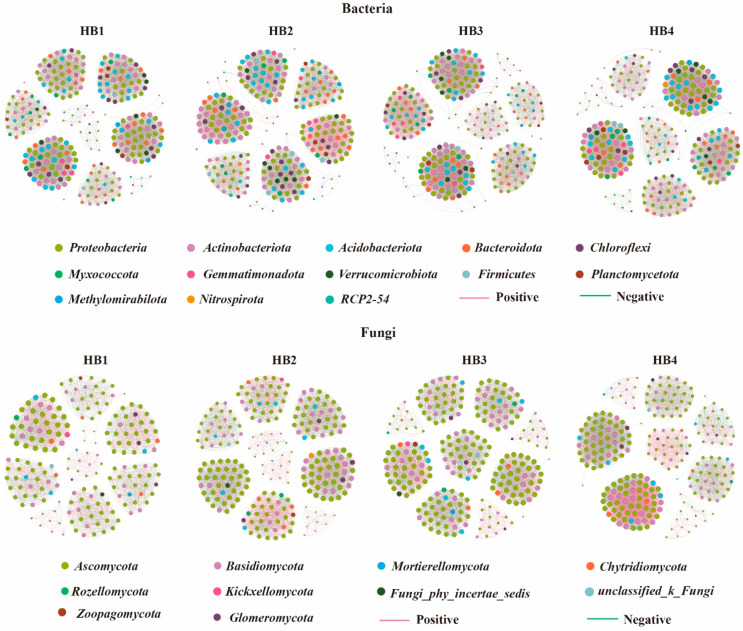
Co-occurrence network analysis of soil microbial communities. Note: Node size represents the degree (number of connected nodes). Red edges indicate positive correlations, while green edges denote negative correlations. Nodes are color-coded based on taxonomic classification (e.g., phylum).

**Figure 7 biology-14-01126-f007:**
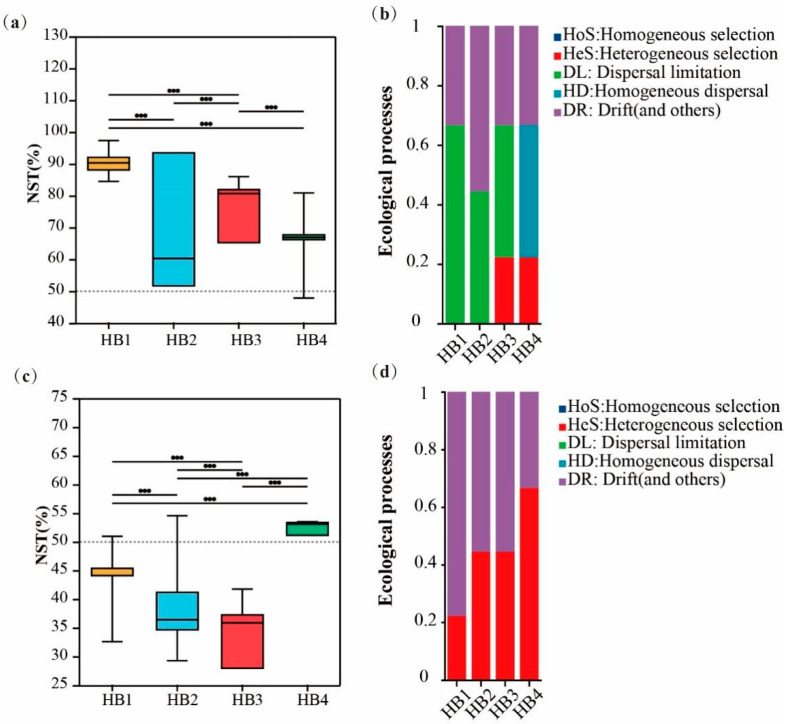
Assembly mechanism of microbial communities in the rhizosphere soil of *Hylodesmum podocarpum* at different altitude gradients. (**a**,**b**) represent bacterial community assembly mechanism; (**c**,**d**) represent fungal community assembly mechanism.

**Figure 8 biology-14-01126-f008:**
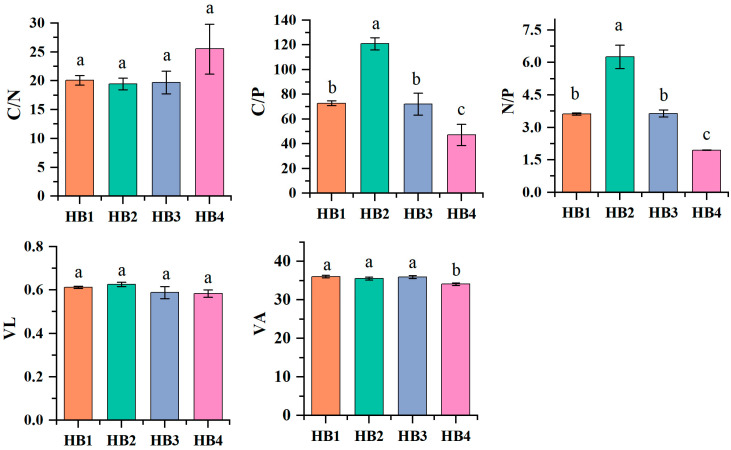
Chemical and enzymatic measurement characteristics of the rhizosphere soil of *Hylodesmum podocarpum* at different altitude gradients. Different lowercase letters denote significant differences among elevations (*p* < 0.05).

**Figure 9 biology-14-01126-f009:**
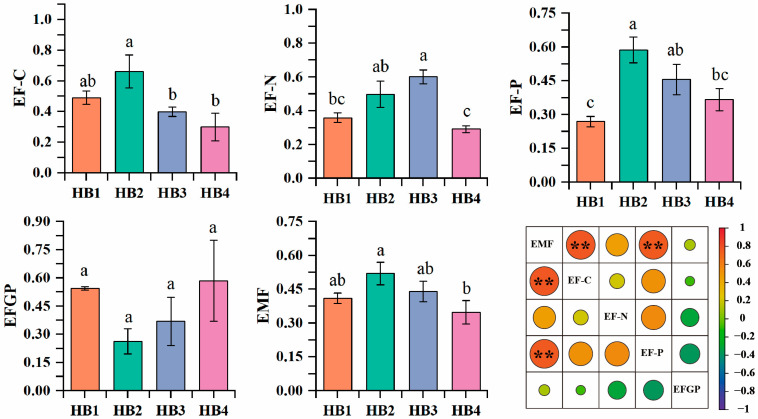
Multifunction of the rhizosphere soil ecosystem of *Hylodesmum podocarpum* at different altitude gradients. Different lowercase letters denote significant differences among elevations (*p* < 0.05). Significance levels: ** (0.001 < *p* < 0.05).

**Figure 10 biology-14-01126-f010:**
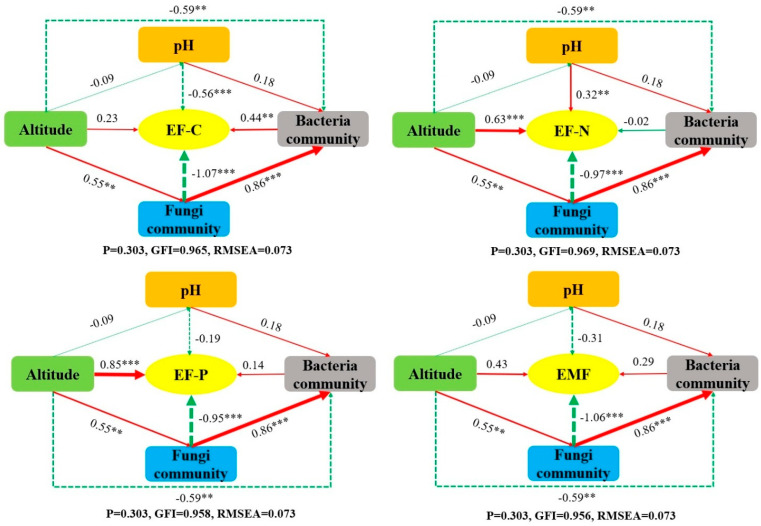
Structural equation models of the effect of altitude gradient on ecosystem function. Significance levels: ** (0.001 < *p* < 0.05); *** (*p* < 0.001).

**Figure 11 biology-14-01126-f011:**
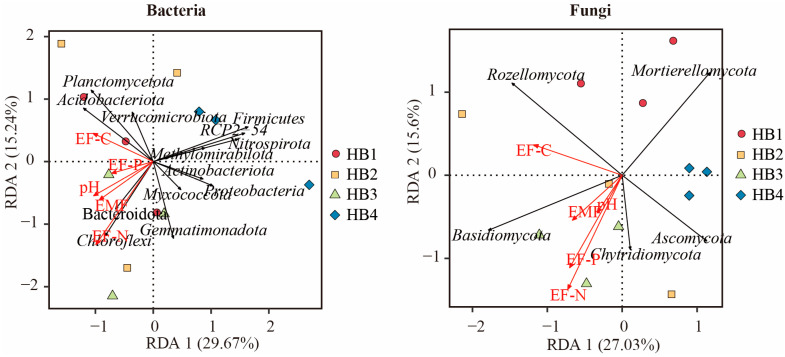
Link between phylum-level species and soil function.

**Table 1 biology-14-01126-t001:** Topological parameters of soil microbial co-occurrence networks.

Topological	Bacteria				Fungi			
Parameters	HB1	HB2	HB3	HB4	HB1	HB2	HB3	HB4
Nodes	272	274	273	272	191	251	259	245
Links	5138	5242	5465	5422	2355	3851	4085	4581
Positive links %	49.77%	51.77%	53.36%	55.91%	57.54%	64.37%	54.96%	55.65%
Negative links%	50.23%	48.23%	46.64%	44.09%	42.46%	35.63%	45.04%	44.35%
Average degree	37.779	38.263	40.073	39.868	24.66	31.437	31.544	36.502
Graph density	0.139	0.14	0.147	0.147	0.13	0.129	0.122	0.146
Modularity	0.811	0.823	0.768	0.776	0.831	0.837	0.849	0.751

## Data Availability

The sequence data associated with this project have been deposited in the NCBI database under accession number PRJNA1291956.

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
