# Peer review of "Mechanisms of Rhizosphere Microbial Regulation on Ecosystem Multifunctionality Driven by Altitudinal Gradients in Hylodesmum podocarpum"

_biology, 2025, doi:10.3390/biology14091126_

Round 1
Reviewer 1 Report
Comments and Suggestions for Authors
I have reviewed the Paper titled "Mechanisms of rhizosphere microbial regulation on ecosystem 2 multifunctionality driven by altitudinal gradients in Podocarpium podocarpum” and my comments are appended below:
- There is serious issue with the accepted name of the species, kindly check the name with POWO website Hylodesmum podocarpum (DC.) H.Ohashi & R.R.Mill | Plants of the World Online | Kew Science. If the name is changed, I suggest to modify throughout the manuscript.
- Check line no. 16
- There is issue with spacing throughout the manuscript
- The introduction section need improvement, specially paragraphs 2nd and 3rd should be merged and compressed only the important point pertaining to study.
- Include the climate data (temperature, rainfall, humidity) of the study area under section 2.1. Also if possible, include map of the study area
- Check line no 114, what authors what to say P. wild P. podocarpum, clearify clearly
- Line 117, The experiment began on July 6, 2024, why only 1 year data, I suggest please include atleast 2 year data
- Presentation of results is Ok
- Discussion need improvement
- Check line 415…t microbial and 435…………..4.2 The effect of altitude gradient on the multifunctionality of the soil ecosystem of . podocarpum
Similarly there are many minor mistakes that need to be corrected.
I suggest it for major revision
Author Response
Dear Editor:
Thanks for your letter and for reviewer's comments concern our manuscript entitled “Mechanisms of rhizosphere microbial regulation on ecosystem multifunctionality driven by altitudinal gradients in Podocarpium podocarpum” (Manuscript ID: 3791688). Those comments are valuable and helpful for revising and improving our paper. We have studied all comments carefully and have made conscientious correction. Revised portion are marked in blue in the paper. The main corrections in the paper and the responds to the reviewer comments are as flowing.
Reviewer 1
I have reviewed the Paper titled "Mechanisms of rhizosphere microbial regulation on ecosystem 2 multifunctionality driven by altitudinal gradients in Podocarpium podocarpum” and my comments are appended below:
- There is serious issue with the accepted name of the species, kindly check the name with POWO website Hylodesmum podocarpum (DC.) H.Ohashi & R.R.Mill | Plants of the World Online | Kew Science. If the name is changed, I suggest to modify throughout the manuscript.
Response: We sincerely thank the reviewer for pointing out the issue with the accepted scientific name of the species. Following your suggestion, we have checked the Plants of the World Online (POWO) database, which indicates that the currently accepted name is Hylodesmum podocarpum (DC.) H.Ohashi & R.R.Mill. In accordance with this authoritative source, we have revised the species name throughout the manuscript, including the title, abstract, main text, figures, tables, and references, to ensure consistency and accuracy.
2.Check line no. 16
Response: Thank you for directing our attention to line 16. We have carefully reviewed this line and revised it for clarity and accuracy.
3.There is issue with spacing throughout the manuscript
Response: We appreciate the reviewer’s observation regarding spacing inconsistencies. We have thoroughly checked the entire manuscript and corrected all spacing issues, including spaces between words, after punctuation marks, and between numbers and units, to ensure formatting is consistent throughout the text.
4.The introduction section need improvement, specially paragraphs 2nd and 3rd should be merged and compressed only the important point pertaining to study.
Response: Thank you for the constructive suggestion. We have revised the introduction section accordingly. Specifically, the 2nd and 3rd paragraphs have been merged and condensed to highlight only the most relevant background information directly related to the objectives of this study. The revised text can be found in lines 50–73 of the revised manuscript.
5.Include the climate data (temperature, rainfall, humidity) of the study area under section 2.1. Also if possible, include map of the study area.
Response: We appreciate the reviewer’s valuable suggestion. We have added detailed climate data for the study area, including mean annual temperature, mean annual precipitation, and mean relative humidity, in Section 2.1 of the revised manuscript. These additions can be found in lines 95–97 of the revised manuscript. Besides, We have added the test overview diagram in Figure 1.
6.Check line no 114, what authors what to say P. wild P. podocarpum, clearify clearly
Response: We thank the reviewer for pointing out this unclear expression. We have carefully revised line 114 to clearly indicate that it refers to “wild Hylodesmum podocarpum populations” in the study area. The corrected wording can be found in the revised manuscript at line 101-103.
7.Line 117, The experiment began on July 6, 2024, why only 1 year data, I suggest please include atleast 2 year data
Response: We sincerely thank the reviewer for the careful review of our manuscript and for the insightful comment regarding the duration of the data. We fully agree that including at least two years of data would enhance the robustness of the conclusions. Here, we would like to clarify that the current manuscript reports the first-year results from our long-term research project on Hylodesmum podocarpum. The experiment indeed began on July 6, 2024, and at the time of manuscript preparation, we had complete data for the first year. Field monitoring and data collection have been ongoing without interruption, and we have already accumulated and are continuing to collect the second-year data.
We chose to publish the first-year findings at this stage because these results have important standalone value and provide a solid foundation for subsequent long-term analyses. We acknowledge that a one-year dataset has limitations, particularly in assessing long-term stability and interannual variation. Addressing these limitations is a core component of our project design, and we plan to conduct a more in-depth multi-year trend analysis once at least four years of data have been collected.
In the revised manuscript, we have clarified in the “Limitations” section that the present study represents the first phase of an ongoing project, and we have explicitly stated both the limitations of the one-year dataset and our plan for continued long-term data collection (lines 504-511).
9.Discussion need improvement
Response: We sincerely thank the reviewer for this valuable suggestion. We have revised and improved the Discussion section to enhance clarity, coherence, and depth. The revisions can be found in lines 391–403, 415–426, 431–432, and 504–511 of the revised manuscript.
10Check line 415…t microbial and 435…………..4.2 The effect of altitude gradient on the multifunctionality of the soil ecosystem of . podocarpum
Response: We thank the reviewer for pointing out these typographical and formatting issues. We have carefully reviewed and corrected the errors in both lines. Specifically, the incomplete phrase “…t microbial” at line 415 has been corrected to its full intended wording, and the species name in the section title at line 435 has been revised to read “Hylodesmum podocarpum” in accordance with the accepted nomenclature. These changes can be found in the revised manuscript at lines 430 and 450.
11Similarly there are many minor mistakes that need to be corrected.
Response: We appreciate the reviewer’s careful reading of the manuscript. We have conducted a thorough revision of the entire text to identify and correct all minor errors, including typographical mistakes, inconsistent formatting, and grammatical issues. These corrections have been made throughout the revised manuscript to improve overall clarity and readability.
Reviewer 2 Report
Comments and Suggestions for Authors
This study provides a comprehensive investigation into how altitudinal gradients modulate rhizosphere microbial regulation of ecosystem multifunctionality in Podocarpium podocarpum. The experimental design is robust, methodologies are state-of-the-art, and findings offer novel insights into microbial ecology. However, methodological transparency, interpretation of contradictory results, and mechanistic depth require improvement.
- The keyword "Community building" should be changed to "Community assembly".
- In line 46, change 'serving' to 'serves'.
- In line 46, change 'Increasing evidence recognized that...' to 'Increasing evidence recognizes that.'
- Section 2.2 mentions collecting plant samples to measure nitrogen, phosphorus, and potassium content, while section 2.3 only describes soil carbon, nitrogen, and phosphorus determination methods. The methods for determining stem and leaf carbon, nitrogen, and phosphorus content are completely missing.
- Reference 27 does not use the enzyme activity measured by the authors to calculate the enzyme activity ratio.
- Section 3.2 claims that 'the Shannon index and Chao1 index of bacteria and fungi increase with altitude,' but Figure 4 shows that the fungal Shannon index is lowest at HB3 (1597m), contradicting the statement of 'continuous increase.
- Lines 284-285, the number of network connections (4735) does not match the data in Table 1.
- The error in VA in Figure 7 is too large, raising doubts about the reliability of the results.
- The discussion section mentions that HB3 (1597m) shows the lowest fungal diversity (Figure 4) and the most complex bacterial network (Table 1), but it does not analyze the reasons for these findings.
Author Response
Dear Editor:
Thanks for your letter and for reviewer's comments concern our manuscript entitled “Mechanisms of rhizosphere microbial regulation on ecosystem multifunctionality driven by altitudinal gradients in Podocarpium podocarpum” (Manuscript ID: 3791688). Those comments are valuable and helpful for revising and improving our paper. We have studied all comments carefully and have made conscientious correction. Revised portion are marked in blue in the paper. The main corrections in the paper and the responds to the reviewer comments are as flowing.
Reviewer 2
This study provides a comprehensive investigation into how altitudinal gradients modulate rhizosphere microbial regulation of ecosystem multifunctionality in Podocarpium podocarpum. The experimental design is robust, methodologies are state-of-the-art, and findings offer novel insights into microbial ecology. However, methodological transparency, interpretation of contradictory results, and mechanistic depth require improvement.
1.The keyword "Community building" should be changed to "Community assembly".
Response: We thank the reviewer for this suggestion. We have replaced the keyword “Community building” with “Community assembly” in the revised manuscript to ensure accuracy and alignment with standard ecological terminology.
2.In line 46, change 'serving' to 'serves'.
Response: Thank you for pointing out this grammatical issue. We have revised the wording in line 46 from “serving” to “serves” in the revised manuscript to improve grammatical accuracy.
3.In line 46, change 'Increasing evidence recognized that...' to 'Increasing evidence recognizes that.'
Response: We appreciate the reviewer’s careful attention to wording. The phrase in line 46 has been revised from “Increasing evidence recognized that...” to “Increasing evidence recognizes that.” to ensure correct tense and grammatical accuracy.
4.Section 2.2 mentions collecting plant samples to measure nitrogen, phosphorus, and potassium content, while section 2.3 only describes soil carbon, nitrogen, and phosphorus determination methods. The methods for determining stem and leaf carbon, nitrogen, and phosphorus content are completely missing.
Response: We sincerely thank the reviewer for this valuable comment. We have added a detailed description of the methods used to determine the carbon, nitrogen, and phosphorus content of stems and leaves in the revised manuscript. The additions can be found in lines 126–134 of the revised manuscript.
5.Reference 27 does not use the enzyme activity measured by the authors to calculate the enzyme activity ratio.
Response: We sincerely thank the reviewer for this valuable comment. The C/N, C/P, and N/P ratios in our study were calculated using soil total carbon, total nitrogen, and total phosphorus contents. The vector length (VL) and vector angle (VA) were calculated based on enzyme activity data. In addition, we applied a logarithmic transformation to the VL and VA calculations to reduce potential error. These details have been clarified in the revised manuscript (lines 322–325).
6.Section 3.2 claims that 'the Shannon index and Chao1 index of bacteria and fungi increase with altitude,' but Figure 4 shows that the fungal Shannon index is lowest at HB3 (1597m), contradicting the statement of 'continuous increase.
Response: We thank the reviewer for pointing out this inconsistency. Upon reviewing the data, we confirmed that bacterial and fungal diversity and richness vary significantly among treatments rather than showing a simple continuous increase with altitude. Specifically, for bacteria, the Shannon and Chao1 indices were lower in HB3 than in the other treatments and highest in HB4. For fungi, the Shannon and Chao1 indices were highest in HB1 and lowest in HB2. Overall, microbial diversity and richness were highest in HB4 and lowest in HB3 (Figure 4). We have revised the statement in Section 3.2 to accurately reflect these patterns, removing the “continuous increase” description. The revision can be found in lines 255-259 of the revised manuscript.
7Lines 284-285, the number of network connections (4735) does not match the data in Table 1.
Response: We sincerely thank the reviewer for this valuable comment. We have corrected the number of network connections to ensure consistency with Table 1. The revision can be found in lines 271–275 of the revised manuscript.
8The error in VA in Figure 7 is too large, raising doubts about the reliability of the results.
Response: We sincerely thank the reviewer for this valuable comment. In calculating VL and VA, we applied a logarithmic transformation to reduce potential errors. This detail has been clarified in the revised manuscript (lines 322–325).
9The discussion section mentions that HB3 (1597m) shows the lowest fungal diversity (Figure 4) and the most complex bacterial network (Table 1), but it does not analyze the reasons for these findings.
Response: We thank the reviewer for this constructive comment. In the revised Discussion section, we have added a detailed explanation of these results. These additions can be found in lines 391–403 and 415–426 of the revised manuscript.
Round 2
Reviewer 1 Report
Comments and Suggestions for Authors
Author have improved the manuscript and can be accepted for publication
Reviewer 2 Report
Comments and Suggestions for Authors
This study systematically explores the regulatory mechanisms of Hylodesmum podocarpum rhizosphere microbial communities on ecosystem multifunctionality through an altitudinal gradient experiment. The topic of the paper has ecological significance, with rigorous experimental design and advanced data analysis methods (such as high-throughput sequencing, molecular ecological networks, null models, etc.). The conclusions provide an important microbiological perspective for the restoration of mountain ecosystems. The revised version has been improved based on the first round of peer review comments.